# Colonisation with extended spectrum beta-lactamase-producing and carbapenem-resistant Enterobacterales in children admitted to a paediatric referral hospital in South Africa

**Babatunde O. Ogunbosi**[1,2]*, **Clinton Moodley**[3,4], **Preneshni Naicker**[3,4], **James Nuttall**[1,2], **Colleen Bamford**[3,4], **Brian Eley**[1,2]

1 Paediatric Infectious Diseases Unit, Red Cross War Memorial Children's Hospital, Cape Town, South Africa, 2 Department of Paediatrics and Child Health, University of Cape Town, Cape Town, South Africa, 3 National Health Laboratory Service, Groote Schuur Hospital, Cape Town, South Africa, 4 Division of Medical Microbiology, University of Cape Town, Cape Town, South Africa

* tundeogunbosi@yahoo.com

**Data Availability Statement:** All relevant data are within the paper and its Supporting Information files.

## Abstract

### Introduction

There are few studies describing colonisation with extended spectrum beta-lactamase-producing Enterobacterales (ESBL-PE) and carbapenem-resistant Enterobacterales (CRE) among children in sub-Saharan Africa. Colonisation often precedes infection and multi-drug-resistant Enterobacterales are important causes of invasive infection.

### Methods

In this prospective cross-sectional study, conducted between April and June 2017, 200 children in a tertiary academic hospital were screened by rectal swab for EBSL-PE and CRE. The resistance-conferring genes were identified using polymerase chain reaction technology. Risk factors for colonisation were also evaluated.

### Results

Overall, 48% (96/200) of the children were colonised with at least one ESBL-PE, 8.3% (8/96) of these with 2 ESBL-PE, and one other child was colonised with a CRE (0.5% (1/200)). Common colonising ESBL-PE were *Klebsiella pneumoniae* (62.5%, 65/104) and *Escherichia coli* (34.6%, 36/104). The most frequent ESBL-conferring gene was blaCTX-M in 95% (76/80) of the isolates. No resistance- conferring gene was identified in the CRE isolate (*Enterobacter cloacae*). Most of the *Klebsiella pneumoniae* isolates were susceptible to piperacillin/tazobactam (86.2%) and amikacin (63.9%). Similarly, 94.4% and 97.2% of the *Escherichia coli* isolates were susceptible to piperacillin/tazobactam and amikacin, respectively. Hospitalisation for more than 7 days before study enrolment was associated with ESBL-PE colonisation.

**Funding:** 1. Initials: BOO Grant Number: None Full name of Funder: Departmental Research Award, Department of Paediatrics and Child Health, University of Cape Town URL of Funder: http://www.paediatrics.uct.ac.za/departmental-funding Did the sponsors or funders play any role in the study design, data collection and analysis, decision to publish, or preparation of the manuscript? NO - The funders had no role in study design, data collection and analysis, decision to publish, or preparation of the manuscript. 2. Initials: BOO Grant Number: None Full name of Funder: African Paediatric Fellowship Programme, Department of Paediatrics and Child Health, University of Cape Town URL of Funder: http://www.paediatrics.uct.ac.za/scah/apfp Did the sponsors or funders play any role in the study design, data collection and analysis, decision to publish, or preparation of the manuscript? NO - The funders had no role in study design, data collection and analysis, decision to publish, or preparation of the manuscript. 3. Initials: BE Grant Number: None Full name of Funder: Department of Paediatrics and Child Health, University of Cape Town URL of Funder: http://www.paediatrics.uct.ac.za/ Did the sponsors or funders play any role in the study design, data collection and analysis, decision to publish, or preparation of the manuscript? NO - The funders had no role in study design, data collection and analysis, decision to publish, or preparation of the manuscript.

**Competing interests:** The authors have declared that no competing interests exist.

## Conclusion

Approximately half of the hospitalised children in this study were colonised with ESBL-PE. This highlights the need for improved infection prevention and control practices to limit the dissemination of these microorganisms.

## Introduction

Infection with multidrug-resistant organisms is a threat to global health [1, 2]. Notable are the emergence of extended spectrum beta-lactamase-producing Enterobacterales (ESBL-PE) and carbapenem-resistant Enterobacterales (CRE) [3]. These organisms are among pathogens categorised as critical on the World Health Organization list of priority antibiotic resistant pathogens [3]. These organisms are notorious for causing healthcare-associated infections, and there are growing concerns about their increasing contribution to community-acquired infections [4]. Invasive infection with CRE is associated with high mortality, high cost of management, few therapeutic options and the absence of effective consensus treatment guidelines [5].

Colonisation refers to the presence of a microorganism on a body surface such as the gastrointestinal tract without causing disease. Colonisation is distinct from infection in which there is invasion of bodily tissues by a disease-causing microorganism, or contamination caused by the accidental introduction of a microorganism during the course of sample collection, transport or processing. Bacterial colonisation often precedes infection, however reports of ESBL-PE and CRE colonisation progressing to infection in children are rare [6, 7]. Most paediatric studies on ESBL-PE colonisation are in neonatal intensive care units (NICUs) with reported prevalence ranging from 4.3% to 75% [8–10]. Outside the neonatal age, ESBL-PE colonisation among hospitalised paediatric patients have recorded prevalence between 18.5% and 57.1% [11–13] while carriage in community settings is lower, ranging from 0.1% to 12.4% [14–18]. In some settings, community carriage has increased over time. For example, surveillance studies in Bolivia and Peru, over a decade, show a steady rise from 0.1% in 2002 to 12.4% in 2011 [14, 18, 19]. Community carriage may be a reservoir for the development of community-acquired ESBL-PE infections, particularly *Escherichia coli* infections [4].

Factors which predispose to ESBL-PE colonisation in the neonatal period include prior carbapenem, penicillin and aminoglycoside exposure, prolonged hospitalisation, early onset pneumonia, prolonged antibiotic use, formula milk feeds, and having an ESBL-PE colonised mother [8, 9, 10, 20–24]. Risk factors for ESBL-PE colonisation in older children include mechanical ventilation, prolonged hospital stay, prior antibiotic use and prior hospitalisation [11, 12]. However, ESBL-producing *E. coli* carriage documented among children in a very remote community in Senegal, where prior antibiotic use was most unlikely, suggesting that other factors may play a role in promoting colonisation [15]. Conversely, breastfeeding confers protection against ESBL-PE colonisation in the newborn period [10, 25]. In early studies, ESBL-PE colonisation was associated with the sulphydryl variable (SHV) enzyme. More recent studies suggest that the cefotaxime-hydrolyzing beta-lactamase, (CTX-M) enzyme now predominates because of superior transmission efficiency. [4, 9, 21, 22, 25, 26].

Studies in children have reported CRE colonisation rates of 0.5% to 29.5% [27–39]. All the studies on CRE colonisation have documented *Klebsiella pneumoniae* colonisation [30–32, 34, 36, 39] while two studies also documented colonisation with *Serratia* spp., *E. coli* and *Enterobacter* spp. [30, 39]. Though there are likely due to geographical variations, one or more carbapenemases have been reported on all continents [30–32, 34, 39]. Risk factors associated with

CRE colonisation include carbapenem exposure, transfer-in from other health facilities, aminoglycoside exposure, surgical procedures, urinary catheterisation, nasogastric intubation and prolonged antibiotic administration [31, 34, 36, 39]. Infection risk among children previously colonised with CRE have ranged from 3.4 to 28.2% [30, 34, 36, 39]. Risk of children colonised with carbapenem-resistant *K. pneumoniae* progressing to infection is increased by existing metabolic disease, previous carbapenem use, neutropaenia and prior surgical procedures [36].

Despite the global distribution of these organisms and the association of their emergence due to widespread and indiscriminate use of broad-spectrum antibiotics, little is known about the epidemiology of ESBL-PE and CRE colonisation and infection on the African continent, especially in children. There have been reports of ESBL-PE infections in South Africa from the early 1990s, and some more recent studies describing ESBL-PE infections, but few colonisation studies [40–43]. One of the first reports of CRE infection and colonisation in children in South Africa described the treatment and outcome of a case series of invasive CRE infections [44]. To address the dearth of ESBL-PE and CRE colonisation research in children in South Africa, we describe ESBL-PE and CRE colonisation at a children's hospital including prevalence, factors associated with ESBL-PE colonisation, and the distribution of ESBL genes among ESBL-PE isolates.

## Methods

### Study design and setting

This prospective, cross-sectional study was conducted at Red Cross War Memorial Children's Hospital (RCWMCH) in Cape Town, South Africa, a 273-bedded tertiary hospital affiliated to the University of Cape Town academic complex and dedicated to the care of children aged 13 years and below. The hospital is a referral centre for sick children from the Western Cape Province, but also receives referrals from surrounding provinces.

### Enrolment and sampling

Recruitment took place in four wards at RCWMCH, including two general medical wards and two surgical wards. All children admitted to the four wards were eligible for enrolment, except those who experienced colonisation or infection with ESBL-PE and/or CRE in the previous 1-year period.

A systematic sampling approach was employed for selecting study participants. From the daily ward list of children resident in the four wards, every third patient was approached for enrolment. If the inclusion criteria were not met, the next patient on the ward list was approached for enrolment. Enrolment took place from Monday to Friday. Between 3 and 8 participants were enrolled on any one day. Enrolment alternated between the medical and surgical participants to ensure that each participant type constituted a minimum of 45% of all enrolled study participants. Participant enrolment was completed between 3 April and 7 June 2017.

### Data collection

Demographic and clinical information of the enrolled participants was entered on a study-specific structured data sheet. The information included age at enrolment, gender, source of admission–either from a health care facility or convalescence home, current diagnosis, surgical procedures, and intensive care unit (ICU) treatment in the current admission or the preceding 12 months. Information on antibiotic use was also collected such as the type and duration of antibiotic therapy in the 12 months preceding this admission or in the current admission. Information on out of hospital antibiotics exposure was mainly as reported by the care giver,

while information on in-hospital antibiotics used was collected from the participant's clinical records.

## Stool specimen collection

A soft tipped TransystemTM sterile transport swab (COPAN Italia S.a.A via Perotti 10, 25125 Brescia Italy) was used to collect rectal stool specimens in a quiet, comfortable room. Swabbing was performed by an experienced research nurse assisted by a chaperone and usually in the presence of the child's parent or legal guardian. All measures were taken to minimise participant discomfort. The specimens were transported to the National Health Laboratory Service (NHLS) medical microbiology laboratory, Groote Schuur Hospital (GSH), Cape Town, for processing within 6 hours of collection.

## Microbiological procedures

All microbiological procedures were performed at the medical microbiology laboratory, GSH, located 4.3 km from RCWMCH. Stool specimens were plated onto ChromID ESBL media (bioMérieux, Marcy l'Etoile, France) and incubated at 37˚C for 24 hours. Suspected colonies were subcultured onto blood agar plates for pure growth. Purified isolates were identified using the Vitek 2Ⓡ Gram negative card with susceptibility testing performed using the AST GN-N255 card (bioMérieux, Marcy l'Etoile, France). Where necessary, this was supplemented with E-test (bioMérieux, Marcy l'Etoile, France) gradient diffusion to confirm minimal inhibitory concentrations (MICs) of ertapenem, imipenem and meropenem. MICs were interpreted according to the 2017 Clinical and Laboratory Standards Institute (CLSI) guidelines [45] while ESBLs and CRE were reported based on the Advanced Expert System (AES) interpretation of the Vitek 2Ⓡ system. The susceptibility of Enterobacterales isolated was evaluated for the following antibiotics: ampicillin, co-amoxiclav, piperacillin-tazobactam, cefuroxime, cefoxitin, ceftriaxone, ceftazidime, cefepime, ertapenem, meropenem, imipenem, ciprofloxacin, gentamicin, amikacin, cotrimoxazole, tigecycline and colistin. All isolates were stored at -70˚C on microbeads for further molecular testing for EBSL and carbapenemase-producing genes.

## Molecular testing for common ESBL and carbapenemase genes

Stored isolates were grown overnight (18–24 hours) on McConkey agar plates, before sub-culturing onto 2% Blood agar. A loop-full of each culture was suspended in 750 μl BashingBead™ Buffer and pre-lysed on the Tissue Lyser in a ZR BashingBead™ Lysis Tube (Zymo Research Corporation), at 50Hz for 5 min. The Lysis tube was centrifuged at 10 000 rpm, and 200 μl of the supernatant extracted and purified using the QIAsymphony DSP Virus/Pathogen Kit (Qiagen), according to the manufacturer's recommendations.

Using gene-specific primers, selected commonly encountered ESBL and carbapenemase genes were detected and amplified by PCR reaction. The primers used are shown in Table in S1 File and detailed methods are detailed in a recent publication [46]. The amplification products were analysed by agarose gel electrophoresis with 1.5%, and visualised with ethidium bromide and using ultraviolet light. Positive amplicons were confirmed with Sanger sequencing, and BLAST analysis.

## Definitions

**HIV-exposed but uninfected child.** A child <18 months old in whom a positive HIV serological test was documented in either the mother or the child, but the HIV DNA PCR test was negative in the child who was not on antiretroviral therapy (ART).

**HIV infection.** A positive HIV DNA PCR result confirmed by either a HIV RNA PCR or repeat HIV DNA PCR test in any child < 18 months old, or 2 positive serological test results (HIV ELISA or HIV Rapid test) or a positive HIV DNA PCR result confirmed by either a HIV RNA PCR or repeat HIV DNA PCR test in a child > 18 months old were considered HIV-infected [47].

**Extended Spectrum Beta-Lactamase Producing Enterobacterales (ESBL-PE).** Enterobacterales were categorised as ESBL-PE according to the Vitek 2® Advanced Expert System (AES) interpretation of the AST GN-N255 card. This categorisation is based solely on the pattern of susceptibility and resistance to different cephalosporins as the card lacks wells containing cephalosporins combined with a beta-lactamase inhibitor [48].

**Carbapenem-Resistant Enterobacterales (CRE).** Enterobacterales which are resistant to any carbapenem antibiotic (minimum inhibitory concentration of $\geq$4 mcg/ml for doripenem, meropenem or imipenem, or $\geq$2 mcg/ml for ertapenem) according to 2017 CLSI breakpoints [45]. Resistance to carbapenems may result from several mechanisms including alteration of outer membrane permeability or the production of carbapenemases. Common carbapenemase genes include $bla_{NDM}$, $bla_{KPC}$, $bla_{GES}$, $bla_{VIM}$, $bla_{OXA-48-like}$ and $bla_{IMP}$ [45].

**Co-morbidity.** An underlying chronic medical condition for which the participant was receiving care at the time of enrolment.

**Major surgery.** An invasive operative procedure in which extensive resection is performed, e.g. a body cavity is entered, organs are removed, or normal anatomy is altered. In general, if a mesenchymal barrier is opened (pleural cavity, peritoneum, meninges), the surgery is considered major [49].

## Statistical analysis

Data were entered in SPSS Statistics Version 24.0 Software (IBM, Armonk, New York, USA) and analysed. Descriptive statistics, for continuous variables, was reported as medians with interquartile ranges or, where applicable as means and standard deviations. Categorical variables were reported as proportions and percentages. The 95% confidence interval (CI) for binomial proportions were estimated for mid-point prevalence estimates. Categorical variables were compared using either the uncorrected chi square test or Fisher's exact test while continuous variables were analysed using the Student's t test or analysis of variance (ANOVA). Non-normally distributed data were compared using Mann-Whitney U test. Two-tail p values <0.05 were considered statistically significant.

Univariate analyses were used to identify potential risk factors associated with rectal colonisation by ESBL-PE. All factors with a p value <0.2 on univariate analysis, and those biologically plausible or reported in literature, were then analysed in a binomial logistic regression model to identify factors independently associated with ESBL-PE colonisation. The binomial logistic regression model was built using a stepwise backward selection. The univariate results were reported using unadjusted odds ratios (ORs) and 95% confidence interval (95% CI), and the logistic regression results expressed as adjusted odds ratios (aORs) and 95% CI.

## Ethical consideration

The study was conducted in accordance with the Helsinki Declaration. The study protocol was approved by the Human Research Ethics Committee, Faculty of Health Sciences, University of Cape Town, reference number: HREC REF: 898/2016. The Research Committee at the RCWMCH also approved the study.

Written informed consent was obtained in the preferred language (English, Afrikaans or isiXhosa) from the parents or legal guardian, and children aged 12 years and above before

enrolment. Children aged between 7 years and above provided written informed assent before being enrolled into the study. The services of translators were employed where necessary during consenting, assenting and enrolment procedures.

Children and their legal guardian were informed of their ESBL-PE and/or CRE colonisation status. When colonisation was documented, the implications for care management, and infection control practices were explained. The result was also provided to the attending physicians and other health care providers involved in their care. Parents or legal guardian of children who had been discharged at the time of receipt of the results were informed telephonically.

## Results

### Study participants

Of a total of 299 children who were selected for enrolment, 99 were excluded for various reasons. Thus, a total of 200 children were enrolled and completed the study. Study participant enrolment, phenotypic and genotypic results are depicted in Fig 1.

### Characteristics of participants

Of the 200 enrolled children, 60.5% were male, the median age was 12 months (range 4 days–7 years and 4 months), and 59.6% (119/200) were less than 24 months of age. The HIV status was known in 153 participants; 3.5% (7/200) were HIV-infected, 19.5% (39/200) were HIV-exposed but uninfected and 53.5% (107/200) were HIV-unexposed. The HIV status in 23.5% (47/200) participants was unknown, as HIV status is not determined routinely in all patients admitted into the hospital. Of the 47 children with unknown HIV status, 89.4% (42/47) were enrolled in the surgical wards. The most frequent primary diagnoses were pneumonia, bone and soft tissue infection and acute diarrhoeal disease (Table 1).

### Prevalence of ESBL-PE and CRE colonisation

Of 200 participants enrolled, 96 were colonized by at least one ESBL-PE, giving an ESBL-PE colonisation prevalence of 48% (95% CI 40.9–55.2%). Of these, 8.3% (8/96) participants were colonised by two different ESBL-PE, prevalence of 4% (95% CI 1.9–7.5%). One additional participant was colonised by a carbapenem-resistant *Enterobacter cloacae (E. cloacae)* giving a CRE colonisation prevalence of 0.5% (95% CI 0.02–2.4%). Of the total 104 ESBL-PE isolates collected from the 96 ESBL-PE colonized participants, *K. pneumoniae* accounted for 62.5% (65/104), *E. coli* 34.6% (36/104), *Klebsiella oxytoca (K. oxytoca)* 1.9% (2/104), and *E. cloacae* 1.0% (1/104). There was no significant difference in the proportions of colonised and non-colonised children with pre-existing co-morbid conditions (Table 2).

Nine percent (18/200) of the enrolled children were transferred-in from another healthcare facility, primarily to the medical wards at RCWMCH, 72.2% (13/18). The median duration of hospital stay at the time of enrolment was 4 days (range 1–64 days) and 30% of the participants had been hospitalised for more than 7 days prior to enrolment (Table 2). About a third of the participants, 32.5% (65/200), had been admitted into other wards in the course of the current admission, aside from the wards in which they were enrolled during the study, and the majority of these children were enrolled in the medical wards, 96.9% (63/65). Of the children admitted into other wards before enrolment, 52.3% (34/65) had been treated in the ICU. In all, 19% (38/200) had been treated in the ICU at RCWMCH either in the current admission or in the preceding 12 months (Table 2).

Overall, 25% (50/200) of the participants reported antibiotic usage in the 12-month period before current hospitalisation, 29.9% (29/97) of colonised children and 20.4% (20/103) of non-colonised children. The most common antibiotics administered during this period were penicillins in

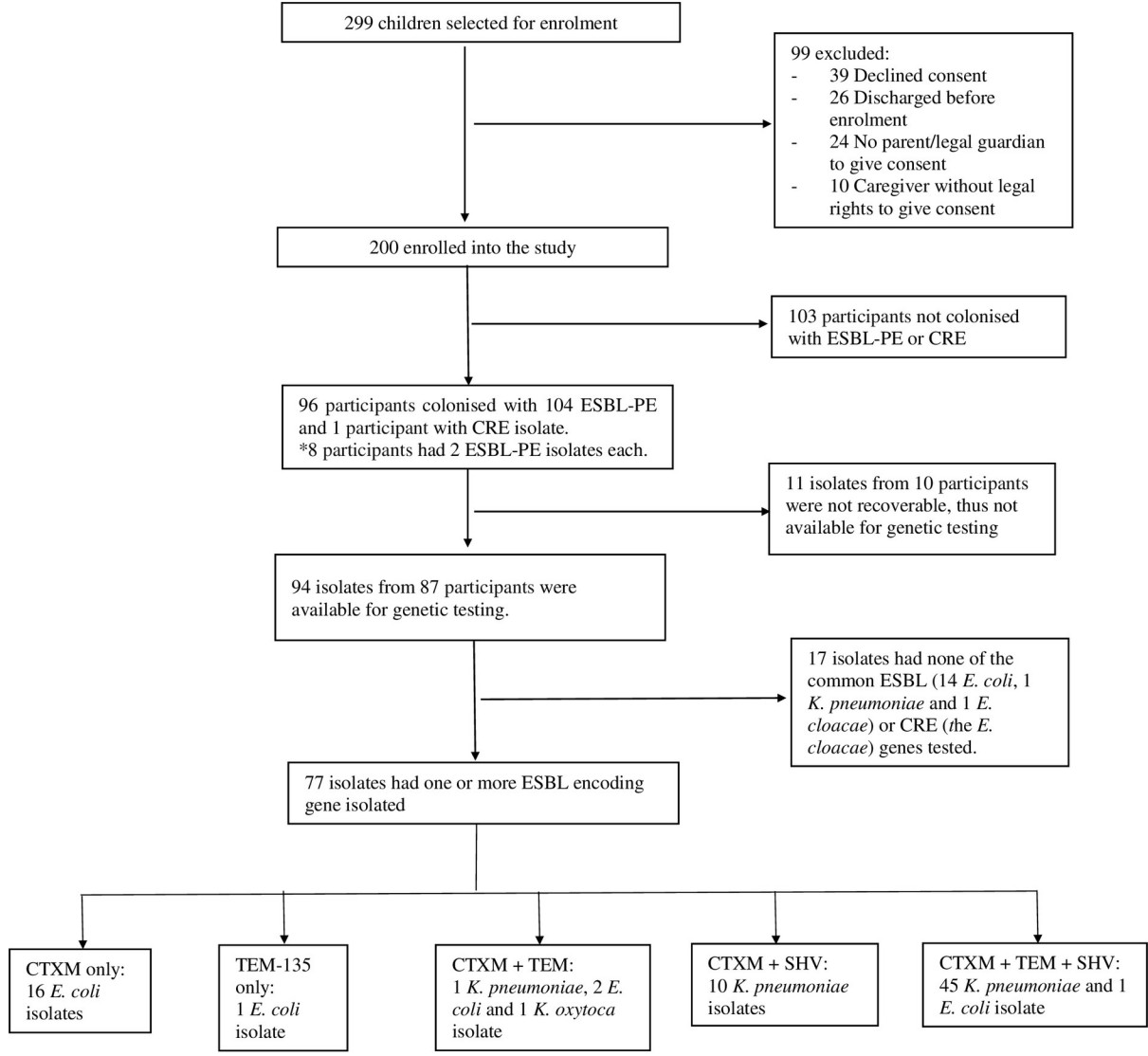

**Fig 1. Flow chart depicting participant enrolment, phenotypic and genotypic testing results.** # *ESBL-PE*, Extended spectrum beta-lactamase producing Enterobacterales, *CRE*, Carbapenem-resistant Enterobacterales, *CTX-M*, Cefotaxime-hydrolyzing beta-lactamase, *TEM*, Temoneira resistance encoding genes, *SHV*, Sulfhydryl variable, *K. pneumoniae—Klebsiella pneumoniae, E. coli—Escherichia coli, K oxytoca—Klebsiella oxytoca, E. cloacae–Enterobacter cloacae.*

15.0% (30/200) of children, while 4.5% (9/200) received a carbapenem. During the current admission, 80.5% (165/200) had received one or more antibiotics, 84.5% (82/97) of colonised children and 76.7% (79/103) of non-colonised children. The antibiotics that these children received during the current period of hospitalisation included penicillins in 55.0% (110/200), aminoglycosides in 32% (64/200), cephalosporins in 19.5% (39/200), beta-lactam/beta-lactam inhibitor combination in 16% (32/200), carbapenems in 9% (18/200), cotrimoxazole in 7.5% (15/200), macrolides in 5.5% (11/200), metronidazole in 4.0% (8/200) and fluoroquinolones in 3.0% (6/200).

## Factors associated with ESBL-PE colonisation

Using univariate analysis, age less than 12 months, admission to a medical ward at the time of study enrolment, hospitalisation for more than 7 days before study enrolment, and nasogastric

**Table 1. Characteristics of the study subjects at the time of enrolment.**

| | Total (N = 200) n (%) | Colonised (N = 97) n (%) | Not colonised (N = 103) n (%) |
|---|---|---|---|
| **Gender** | | | |
| Male | 121 (60.5) | 58 (59.8) | 63 (61.2) |
| Female | 79 (39.5) | 39 (40.2) | 40 (38.8) |
| **Age category** | | | |
| ≤28 days | 9 (4.5) | 6 (6.2) | 3 (2.9) |
| >28 days– 12 months | 89 (44.5) | 50 (51.5) | 39 (3.79) |
| >12 months—<60 months | 47 (23.5) | 20 (20.6) | 27 (26.2) |
| ≥60 months | 55 (27.5) | 21 (21.6) | 34 (33.0) |
| **Median age in months (IQR)** | 12 (2–68) | 7 (2–45) | 18 (4–82) |
| **Median (IQR) days in hospital before enrolment** | 4 (2–9) | 5 (2–12) | 3 (2–7) |
| **HIV status** | | | |
| Infected | 7 (3.5) | 4 (4.1) | 3 (2.9) |
| Exposed, uninfected | 39 (19.5) | 16 (16.5) | 23 (22.3) |
| Unexposed, uninfected | 107 (53.5) | 65 (67.0) | 42 (40.8) |
| Unknown | 47 (23.5) | 12 (2.4) | 35 (34.0) |
| **Primary clinical diagnosis** | | | |
| Pneumonia | 64 (32.0) | 33 (34.0) | 31 (30.1) |
| Bone and soft tissue infection | 20 (10.0) | 7 (7.2) | 13 (12.6) |
| Acute diarrhoeal disease | 11 (5.5) | 8 (8.2) | 3 (2.9) |
| Central nervous system malformation | 10 (5.0) | 6 (6.2) | 4 (3.9) |
| Appendicitis | 8 (4.0) | 1 (1.0) | 7 (6.8) |
| Meningitis | 8 (4.0) | 2 (2.1) | 6 (5.8) |
| Urogenital malformations | 7 (3.5) | 2 (2.1) | 5 (4.9) |
| Hydrocephalus | 7 (3.5) | 0 (0.0) | 7 (6.8) |
| Hirschsprung disease | 5 (2.5) | 2 (2.1) | 3 (2.9) |
| Trauma | 5 (2.5) | 2 (2.1) | 3 (2.9) |
| Neonatal sepsis | 4 (2.0) | 4 (4.1) | 0 (0.0) |
| Inguinal hernia | 4 (2.0) | 2 (2.1) | 2 (1.9) |
| Tuberculosis | 3 (1.5) | 1 (1.0) | 2 (1.9) |
| Solid tumour | 3 (1.5) | 2 (2.1) | 1 (1.0) |
| Bloodstream infection | 2 (1.0) | 0 (0.0) | 2 (1.9) |

intubation at the time of enrolment were significantly associated with ESBL-PE colonisation. However, on binomial logistic regression analysis, only hospitalisation for more than 7 days before study enrolment remained significantly associated with ESBL-PE colonisation (p = 0.013) (Table 2).

## Antibiotic susceptibility profile of the ESBL-PE and CRE isolates

Table 3 summarises the antibiotic susceptibility profiles of the 104 ESBL-PE isolates. All ESBL-PE isolates were resistant to the cephalosporins except for one *E. coli* isolate and one *E. cloacae* isolate which retained susceptibility to cefepime, using Vitek 2. All the ESBL-PE isolates were however susceptible to the carbapenems and colistin. Most of the *K. pneumoniae* (72.3%) were susceptible to piperacillin/tazobactam and to amikacin (86.2%). Similarly, 94.4% and 97.2% of the *E. coli* isolates were susceptible to piperacillin/tazobactam and amikacin, respectively (Table 3). Susceptibility to both piperacillin/tazobactam and amikacin was observed in 73.1% of all isolates, 64.6% and 91.7% of all ESBL-PE *K. pneumoniae* and *E. coli*

**Table 2. Factors associated with extended spectrum beta-lactamase-producing Enterobacterales colonisation determined by univariate analyses and binomial logistic regression.**

| | Colonised N = 96 n (%) | Not colonised N = 104 n (%) | OR (95% CI) | aOR (95% CI) |
|---|---|---|---|---|
| Male gender | 38 (39.6) | 41 (39.4) | 1.01 (0.57–1.78) | |
| Age < 12 months | 55 (57.3) | 43 (41.3) | 1.90 (1.09–3.34) * | 1.36 (0.70–2.65) |
| Transferred to RCWMCH[@] | 10 (10.4) | 9 (8.7) | 1.23 (0.48–3.16) | |
| Previous RCWMCH admission | 23 (24.0) | 15 (14.4) | 1.87 (0.91–3.84) | 1.23 (0.52–2.91) |
| Current admission to medical ward (vs. surgical ward) | 63 (65.6) | 49 (47.1) | 2.14 (1.21–3.79) * | 0.78 (0.35–1.77) |
| Hospitalisation for >7 days before enrolment | 40 (41.7) | 20 (19.2) | 3.0 (1.59–5.66) * | 2.83 (1.40–5.72) * |
| Cardiac co-morbidity | 8 (8.3) | 4 (3.8) | 2.27 (0.66–7.81) | |
| Gastrointestinal tract co-morbidity | 20 (20.9) | 22 (21.2) | 0.98 (0.50–1.94) | |
| Neurological co-morbidity | 17 (17.7) | 16 (15.4) | 1.18 (0.56–2.50) | |
| Concomitant chronic lung disease | 7 (7.3) | 3 (2.9) | 2.65 (0.67–10.55) | 2.67 (0.57–12.59) |
| Concomitant congenital anomaly | 25 (26.0) | 25 (24.0) | 1.11 (0.59–2.11) | |
| Major surgery in current admission | 12 (12.5) | 23 (22.1) | 0.50 (0.24–1.08) | 0.40 (0.15–1.04) |
| Peripheral venous line | 88 (91.7.7) | 87 (83.7) | 2.15 (0.88–5.24) | 1.81 (0.65–5.02) |
| Nasotracheal intubation ± ventilation | 8 (8.3) | 4 (3.8) | 2.27 (0.66–7.81) | |
| PEG feeding | 6 (6.2) | 1 (1.0) | 6.87 (0.81–58.1) | |
| Nasogastric intubation | 38 (39.6) | 24 (23.1) | 2.18 (1.18–4.03) * | 1.60 (0.76–3.39) |
| Decreased level of consciousness | 5 (5.2) | 1 (1.0) | 5.66 (0.65–49.3) | |
| Receiving gastric acid inhibitor therapy | 12 (12.5) | 6 (5.8) | 2.33 (0.84–6.49) | |
| Receiving immunosuppressive therapy | 11 (11.2) | 11 (10.6) | 1.09 (0.45–2.65) | |
| Admission to ICU in 12 months preceding current admission | 6 (6.2) | 2 (1.9) | 3.40 (0.68–17.27) | |
| Admission to ICU during current admission | 19 (19.8) | 15 (14.4) | 1.46 (0.70–3.08) | |
| Antibiotic administration in the 12-month period preceding admission | 29 (30.2) | 21 (20.2) | 1.71 (0.90–3.27) | 1.35 (0.63–2.90) |

*p<0.05, OR Odds ratio, 95% CI 95% confidence interval, aOR adjusted Odds ratio, RCWMCH Red Cross War Memorial Children's Hospital, PEG percutaneous endoscopic gastrostomy, ICU intensive care unit. [@]Tranfered to RCWMCH refers to admitted from another hospital or a convalescent home.

isolates respectively. The CRE *E. cloacae* isolate exhibited intermediate susceptibility to ertapenem on E-test, but was susceptible to imipenem and meropenem.

**Table 3. Antibiotic susceptibility profile of the extended spectrum beta-lactamase-producing Enterobacterales isolates showing proportion susceptible.**

| Antibiotic | Total N = 104 (%) | *Klebsiella pneumoniae* N = 65 n (%) | *Escherichia coli* N = 36 n (%) | *Klebsiella oxytoca* N = 2 n (%) | *Enterobacter cloacae* N = 1 n (%) |
|---|---|---|---|---|---|
| Cotrimoxazole | 10 (9.6) | 5 (7.7) | 4 (11.1) | 0 (0.0) | 1 (100.0) |
| Ampicillin/amoxicillin | 1 (1.0) | 0 (0.0) | 1 (2.8) | 0 (0.0) | 0 (0.0) |
| Amoxicillin plus clavulanic acid | 19 (18.3) | 1 (1.5) | 16 (44.4) | 2 (100.0) | 0 (0.0) |
| Piperacillin plus tazobactam | 83 (79.8) | 47 (72.3) | 34 (94.4) | 2 (100.0) | 0 (0.0) |
| Cefepime | 2 (1.9) | 0 (0.0) | 1 (2.8) | 0 (0.0) | 1 (100.0) |
| Ciprofloxacin | 687 (64.4) | 41 (63.1) | 23 (63.9) | 2 (100.0) | 1 (100.0) |
| Gentamicin | 33 (31.7) | 8 (12.3) | 22 (61.1) | 2 (100.0) | 1 (100.0) |
| Amikacin | 93 (89.4) | 56 (86.2) | 35 (97.2) | 1 (50.0) | 1 (100.0) |
| Tigecycline | 104 (100.0) | 65 (100.0) | 36 (100.0) | 2 (100.0) | 0 (0.0) |

## Genetic testing of ESBL-PE and CRE isolates

Of the 104 ESBL-PE and 1 CRE isolated from 97 colonised children, 94 isolates from 87 participants were available for genetic testing. Eighty ESBL-PE isolates (85.1%) from 73 participants had one or more resistance-conferring genes (Table 4). Of these, 25% (20/80) had only one resistant-conferring gene, 14 (17.5%) had 2 genes, and 46 (57.5%) had 3 genes identified. The most common resistance conferring gene was $bla_{CTX-M}$ found in 95% (76/80) of ESBL-PE i.e. in 98.2% (56/57) of *K. pneumoniae* isolates, 55.9% (19/34) of *E. coli* isolates, and the only *K. oxytoca* (1.3%) isolate tested. The $bla_{TEM}$ and $bla_{SHV}$ genes were found in 66.3% (53/80) and 13.8% (11/80) of ESBL-PE isolates respectively, often in combination with $bla_{CTX-M}$ gene. Three *E. coli* isolates had $bla_{TEM}$ as the only resistance gene, and one *K. pneumoniae* isolate had $bla_{SHV}$ as the only resistance gene. The 3 *E. coli* TEM-only positive amplicons were sequenced, and two were identified as TEM-1 genotypes i.e. narrow-spectrum beta-lactamase, and the remaining one as a TEM-135 genotype i.e. ESBL. The DNA from the SHV-only positive *K. pneumoniae* isolate was not sequenced as blaSHV is intrinsic to *K. pneumoniae* and would not be differentiated using conventional sequencing. For the two ESBL-PE isolates which retained susceptibility to cefepime, no gene was found in the *E. cloacae* isolate, while the *E. coli* isolate had both CTX-M and TEM genes.

The only CRE isolate, an *E. cloacae*, was negative for common CRE conferring genes, namely $bla_{NDM}$, $bla_{KPC}$, $bla_{OXA-48}$ and variants, $bla_{IMP}$, $bla_{VIM}$ and $bla_{GES}$.

## Discussion

In this prospective cross-sectional study, we showed that approximately 50% of children hospitalised in RCWMCH are colonised with ESBL-PE, predominately with *K. pneumoniae* and *E. coli*, findings which are consistent with previous paediatric colonisation studies [11, 12, 13]. The CRE colonisation prevalence was extremely low in our participants, but comparable to previous studies which also reported lower CRE colonisation [30, 34, 50]. Similarly, bloodstream infection (BSI) studies from our hospital have shown that while *K. pneumoniae* and *E. coli* are the predominant Gram-negative aetiological agents, CRE only sporadically caused

**Table 4. Frequency of genes conferring extended spectrum beta-lactam resistance to the colonising extended spectrum beta-lactamase-producing Enterobacterales isolates.**

| ESBL Gene | Extended spectrum beta-lactamase producing Enterobacterales N = 104 | | | | Total |
|---|---|---|---|---|---|
| | *Klebsiella pneumoniae n = 65* | *Escherichia coli n = 36* | *Klebsiella oxytoca n = 2* | *Enterobacter cloacae n = 1* | |
| Not available | 8 | 2 | 1 | 0 | 11 |
| No gene found | 0 | 12 | 0 | 1 | 13 |
| **No EBSL gene confirmed** | | | | | |
| TEM 1 only | 0 | 2 | 0 | 0 | 2 |
| SHV 1 only | 1 | 0 | 0 | 0 | 1 |
| **EBSL gene confirmed** | | | | | |
| TEM-135 only | 0 | 1 | 0 | 0 | 1 |
| CTX-M only | 0 | 16 | 0 | 0 | 16 |
| CTX-M + TEM | 1 | 2 | 1 | 0 | 4 |
| CTX-M + SHV | 10 | 0 | 0 | 0 | 10 |
| TEM + SHV | 0 | 0 | 0 | 0 | 0 |
| CTXM + TEM + SHV | 45 | 1 | 0 | 0 | 46 |

ESBL-PE, extended spectrum beta-lactamase-producing Enterobacteriaceae, ESBL, extended spectrum beta-lactamase, CTX-M, cefotaxime-hydrolyzing beta-lactamase, TEM, Temoneira resistance encoding genes, SHV, sulphydryl variable.

invasive infection [44, 51]. Furthermore, of *K. pneumoniae* and *E. coli* isolates causing BSI at RCWMCH, 83% and 30% are ESBL-PE, respectively [41, 42].

This study was not designed to determine the timing of colonisation of ESBL-PE and CRE and hence we were unable to establish the extent of community versus hospital acquisition. Colonisation was documented in some participants who had only been hospitalised for one day, suggesting that some of the participants may have been colonised outside the hospital environment. However, the risk factor analysis showed that hospitalisation for more than 7 days before enrolment was an independent risk factor for ESBL-PE colonisation. This finding strongly suggests that the healthcare environment is an important site of colonisation and is consistent with previous BSI studies that showed that 95% and 55% of *K. pneumoniae* and *E. coli* BSIs respectively are hospital-acquired or healthcare-associated [41, 42].

The predominant ESBL gene identified was $bla_{CTX-M}$ (95%), often occurring in combination with other ESBL-conferring genes (78.9%). This was notable for the *K. pneumoniae* isolates, most of which carried more than one ESBL-conferring gene. In contrast, most *E. coli* isolates harboured only the $bla_{CTX-M}$ gene. This is in keeping with the global pattern where CTX-M has become the predominant ESBL-conferring gene in resistant *K. pneumoniae* and *E. coli* isolates [4, 21, 26]. It has been suggested that the CTX-M plasmid has adapted to *K. pneumoniae* resulting in better transmission efficiency [4, 26]. Three *E. coli* isolates each harboured a TEM gene only. One isolate had a TEM-135 gene which encodes an enzyme that hydrolyses 3rd generation cephalosporins and thus confers ESBL-PE properties on the isolate [52]. This is unlike the TEM-1 gene found in the other 2 isolates, the first TEM gene which encodes an enzyme that hydrolyses earlier penicillins like ampicillin but not 3rd generation cephalosporins, and therefore does not confer ESBL-PE characteristics on the isolates [53]. Most *K. pneumoniae* isolates have chromosomal SHV genes which encodes enzymes that do not hydrolyse 3rd generation cephalosporins [53]. One of our *K. pneumoniae* isolates was shown to only carry an SHV gene. The ESBL-PE phenotype of the three isolates with TEM-1 only or SHV only genes might be due to hyperproduction of the encoded enzymes along with porin changes, or the presence of other ESBL's or enzymes not detected by this assay.

The antibiotic susceptibility profiles of the ESBL-PE isolates as summarised in Table 4 correlate with published data. High proportions of the *K. pneumoniae* and *E. coli* isolates were susceptible to piperacillin-tazobactam and to amikacin, similar to findings among ESBL-producing bloodstream isolates of *K. pneumoniae* and *E. coli* reported in previous studies from RCWMCH [41]. Despite these findings, a randomised control trial among adults that determined whether piperacillin-tazobactam was as effective as meropenem for treating BSI caused by *K. pneumoniae* or *E. coli* with non-susceptibility to third generation cephalosporins showed that the 30-day mortality was significantly higher in the piperacillin-tazobactam group, suggesting that piperacillin-tazobactam alone may no longer be the recommended definitive therapy in this context [54]. The high proportion of *E. coli* (91.6%) susceptible to both piperacillin-tazobactam and amikacin suggest this may be a consideration in BSI caused by ESBL-PE *E. coli*. This is unlikely to be an option for ESBL-PE *K. pneumoniae* where only 64.6% were susceptible to both piperacillin-tazobactam and amikacin.

## Study strengths and limitations

A strength of this study is that it is one of the first prospective studies to estimate the prevalence of ESBL-PE and CRE carriage in children hospitalised in sub-Saharan Africa. Limited funds for completing the microbiology investigation of the enrolled participants restricted our surveillance to one specimen per participant and prevented us from sampling the participants at multiple time points during hospitalisation, and prevented differentiation between

community and hospital acquisition of ESBL-PE. Our sampling approach may have under-estimated the colonisation rate as multiple rectal swabs may have increased the yield of colo-nised children or serial sampling may have identified additional children who became colo-nised later on in the course of their hospital admission. The sample size of 200 children was not determined scientifically but influenced by the available funding. Thus, the study may have been underpowered to explore risk factors associated with colonisation definitively. Screening for genes conferring extended spectrum beta-lactam resistance was limited to the three commonly occurring resistance genes in our setting. Furthermore, we did not screen the isolates for the presence of Ambler group C beta-lactamases which produce similar antibio-grams to the extended spectrum beta-lactamases. Lastly, relatedness of the ESBL-PE isolates was not evaluated. This may have assisted us to understand the extent of hospital transmission.

Notwithstanding these limitations the study showed that ESBL-PE colonisation prevalence is high in this setting, and prolonged hospitalisation is a risk factor for colonisation, suggesting the need for improved infection control practices.

## Conclusion

The current study extends our understanding of ESBL-PE and CRE colonisation at our institu-tion. Major findings were high ESBL-PE colonisation prevalence, very low CRE colonisation prevalence, prolonged hospitalisation as an independent risk factor for ESBL-PE colonisation, and common resistance genes responsible for conferring extended spectrum beta-lactam resis-tance. Further research, adequately funded to include more participants, is required to provide robust colonization estimates, explore colonisation prevalence changes over time, quantify the extent of community versus hospital acquisition, determine the pattern of acquisition of colo-nisation in the course of admission and relatedness of the isolates, and whether improved infection control practice can moderate colonisation rates. Whole genome sequencing will additionally help interrogate other resistance conferring genes not identified by PCR technol-ogy among the ESBL-PE and CRE isolates, especially among those where no resistance confer-ring gene was found.

## Supporting information

**S1 File. References for ESBL and carbapenemase PCR.**
(DOCX)

**S2 File. Data set of study participants and isolates.**
(SAV)

**S3 File. ESBL and CRE PCR results.**
(XLSX)

## Acknowledgments

We are grateful to Spasina King and Lungiswa Williams who assisted with participant enrol-ment, translation and rectal swab sample collection. Gratitude also to Prof Mary-Ann Davies who provided some statistical support.

## Author Contributions

**Conceptualization:** Babatunde O. Ogunbosi, Preneshni Naicker, James Nuttall, Colleen Bam-ford, Brian Eley.

**Data curation:** Babatunde O. Ogunbosi, Brian Eley.

**Formal analysis:** Babatunde O. Ogunbosi, Brian Eley.

**Funding acquisition:** Babatunde O. Ogunbosi, Brian Eley.

**Investigation:** Babatunde O. Ogunbosi, Clinton Moodley, Preneshni Naicker, Colleen Bamford.

**Methodology:** Babatunde O. Ogunbosi, Clinton Moodley, Preneshni Naicker, James Nuttall, Colleen Bamford, Brian Eley.

**Project administration:** Babatunde O. Ogunbosi, Clinton Moodley, Preneshni Naicker, Colleen Bamford, Brian Eley.

**Resources:** Babatunde O. Ogunbosi, Brian Eley.

**Software:** Babatunde O. Ogunbosi, Brian Eley.

**Supervision:** Babatunde O. Ogunbosi, Brian Eley.

**Validation:** Babatunde O. Ogunbosi, Clinton Moodley, Preneshni Naicker, Colleen Bamford, Brian Eley.

**Visualization:** Babatunde O. Ogunbosi, Brian Eley.

**Writing – original draft:** Babatunde O. Ogunbosi, Brian Eley.

**Writing – review & editing:** Babatunde O. Ogunbosi, Clinton Moodley, Preneshni Naicker, James Nuttall, Colleen Bamford, Brian Eley.

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
