## [Decision Letter · Decision Letter 0]

8 Sep 2020

PONE-D-20-22787

Colonisation with extended spectrum beta-lactamase-producing and carbapenem-resistant Enterobacterales in children admitted to a paediatric referral hospital in South Africa

PLOS ONE

Dear Dr. Ogunbosi,

Thank you for submitting your manuscript to PLOS ONE. After careful consideration, we feel that it has merit but does not fully meet PLOS ONE’s publication criteria as it currently stands. Therefore, we invite you to submit a revised version of the manuscript that addresses the points raised during the review process.

We look forward to receiving your revised manuscript.

Kind regards,

Mehreen Arshad, M.D.

Academic Editor

PLOS ONE

Journal Requirements:

Reviewers' comments:

Reviewer's Responses to Questions

**Comments to the Author**

1. Is the manuscript technically sound, and do the data support the conclusions?

Reviewer #1: Yes

Reviewer #2: Yes

Reviewer #3: Yes

2. Has the statistical analysis been performed appropriately and rigorously? 

Reviewer #1: Yes

Reviewer #2: I Don't Know

Reviewer #3: Yes

3. Have the authors made all data underlying the findings in their manuscript fully available?

Reviewer #1: Yes

Reviewer #2: Yes

Reviewer #3: Yes

4. Is the manuscript presented in an intelligible fashion and written in standard English?

Reviewer #1: Yes

Reviewer #2: Yes

Reviewer #3: Yes

5. Review Comments to the Author

Reviewer #1: The manuscript is well written and the data presented is relevant, especially in the region. I have a few observations.

1. Numbers stated in the abstract should be clearly delineated as either whole numbers or percentages; line 37-38, 8 and 2 referred to.

2.Introduction:line 99-100; The study is small scale even for South Africa alone, so the use of Africa is too generalized. Instead restrict it to South Africa or rather Southern Africa.

3. The sample size used is small, although it was mentioned as a limitation, a wide scale study using different sampling points and sample size would have provided a better data that is robust.

4. Calculation of relative risk among the different sub categorizations such as sex, age , HIV status. e.t.c.

Reviewer #2: This manuscript is well laid out. The topic is important from point of view of epidemiology, antibiotic stewardship and infection control. The language is clear and data is well presented. I cannot comment on statistical analysis as this is not in my expertise.

My only suggestion is for the authors to give a one sentence description of "colonization" as opposed to infection and contamination.

Reviewer #3: The authors of this study set out to characterise the colonisation levels of children in South Africa by beta-lactamase producing Enterobacterales (ESBL-PE) and carbapenem-resistant Enterobacterales (CRE). Samples included in the study were collected prospectively from a cohort of children hospitalised in a tertiary, academically-linked children’s hospital in South Africa. Rectal stool swabs were characterised via growth on media selective for growth of Enterobacterales as well as molecular methods to detect common genes that confer resistance in these organisms. Clinical data for the participants was collected allowing for an investigation of the risk factors associated with Enterobacterales colonisation in this setting. The reported results suggestion that approximately half of children were colonies with ESBL-PE, with Klebsiella pneumoniae and Escherichia coli dominating at the species level. Colonisation with CRE was relatively low in this setting. Colonisation was associated with longer hospitalisation times and less significantly, age at admission, medical award admission at study enrolment and nasogastric intubation.

As made clear by the authors, there are several limitations in this study. The number of samples examined and the extent of both molecular and microbiological characterisation of samples, was limited by available funds and therefore the generalisable conclusions are limited. Nevertheless, the molecular, microbiological and statistical methods are technically sound, and the data both support the authors conclusions and is appropriately available. Moreover, the manuscript is well written and is clear and easy to follow and it’s great that the authors state the limitations up front and in a frank way. Therefore, while the sample size was limited, I would recommend publication of this manuscript. The study could form an important basis for further investigation of nosocomial-associated colonisation of children by ESBL-PE and CRE and thus contributes to this field.

I have made minor suggestions below, which could improve the quality and clarity of the manuscript which could be addressed before publication.

INTRODUCTION

1. Line 54. The word “They” at the start of the sentence could be replaced. Suggestion:

“Theseorganisms are among pathogens categorised as critical on the World Health Organization list of priority antibiotic resistant pathogen…”

2. Line 78 – 79. The authors could explain why ESBL-PE colonisation was associated with the SHV enzyme and reasons for more the predomination of the CTM-X enzyme more recently.

3. Line 125. There is no assessment or investigation around whether or not children were transferred from another healthcare facility or convalescence home – is the sample size big enough to look at this?

4. Line 140. It would be useful to know if samples were processed at the GSH laboratory on the day they were received. And if not, how they were stored and if this could influence the results in any way.

RESULTS

5. Figure 1. It would be better to use the word “participant” rather than “patient” when referring to children enrolled in the study?

6. Line 277. It would be more appropriate to describe the participant characteristics before describing prevalence of ESBL-PE or CRE colonisation.

7. Table 1. In the text, the authors report that 96 children were colonised by either ESBL-PE or CRE, but in the table it’s indicated that 97 children where colonised? This should be corrected.

8. Table 1. It would be useful to know why the HIV category of 23.5% of children is unknown.

9. Line 305. It would be useful to mention in this line, the number of children that had known previous antibiotic use and were colonisation by ESBL-PE.

10. Line 336. Again here the number of colonised children is listed as 97 whereas in other places it’s 96? The manuscript should be checked throughout for consistency.

DISCUSSION

11. It might be interesting to discuss in more detail the kinds of studies that could be conducted to build upon this one and if and how whole-genome sequencing could contribute.

6. PLOS authors have the option to publish the peer review history of their article (what does this mean?). If published, this will include your full peer review and any attached files.

Reviewer #1: No

Reviewer #2: **Yes: **Naseem Salahuddin

Reviewer #3: No

---

## [Author Response · Author response to Decision Letter 0]

6 Oct 2020

Responses to Reviewer’s Comments for PLoS ONE Journal on “Colonisation with extended spectrum beta-lactamase-producing and carbapenem-resistant Enterobacterales in children admitted to a paediatric referral hospital in South Africa” 

Reviewer 1

Comment 1: Numbers stated in the abstract should be clearly delineated as either whole numbers or percentages; line 37-38, 8 and 2 referred to.

Response: The percentage of the 8 participants with 2 ESBLs among all participants colonised with ESBLs has been included.

Correction: Lines 37-38; “Overall, 48% (96/200) of the children were colonised with at least one ESBL-PE, 8.3% (8/96) of these were colonised with 2 ESBL-PE, and one other child was colonised with a CRE (0.5% (1/200)).”

Introduction:

Comment 2: Line 99-100; The study is small scale even for South Africa alone, so the use of Africa is too generalized. Instead restrict it to South Africa or rather Southern Africa.

Response: Suggestion has been incorporated.

Correction: Line 103 – 104; “To address the dearth of ESBL-PE and CRE colonisation research in children in South Africa, we...”

Comment 3: The sample size used is small, although it was mentioned as a limitation, a wide scale study using different sampling points and sample size would have provided a better data that is robust.

Response: The comment is acknowledged and appreciated. Suggestion in this regard has been included in the conclusion.

Correction: Lines 445 - 449; “Further research, adequately funded to include more participants, is required to provide robust colonisation estimates, explore colonisation prevalence changes over time, quantify the extent of community versus hospital acquisition, determine the pattern of acquisition of colonisation in the course of admission and relatedness of the isolates, and whether improved infection control practice can moderate colonisation rates.”

Comment 4: Calculation of relative risk among the different sub categorizations such as sex, age, HIV status. etc.

Response: In response to recent research trends e.g. recently released NEJM statistical guidelines, we elected not to include a statistical comparison of the characteristics of colonised and non-colonised in table 1.

Correction: No correction 

Reviewer 2

Comment 1: My only suggestion is for the authors to give a one sentence description of "colonization" as opposed to infection and contamination.

Response: A brief on this has been included.

Correction: Lines 60-64; “Colonisation refers to the presence of a microorganism on a body surface such as the gastrointestinal tract without causing disease. Colonisation is distinct from infection in which there is invasion of bodily tissues by a disease-causing microorganism, or contamination caused by the accidental introduction of a microorganism during the course of sample collection, transport or processing.”

Reviewer 3

Introduction

Comment 1: Line 54 The word “They” at the start of the sentence could be replaced. Suggestion: “These organisms are among pathogens categorised as critical on the World Health Organization list of priority antibiotic resistant pathogen…”

Response: Suggestion has been incorporated.

Correction: Lines 54 - 56 “These organisms are among pathogens categorised as critical on the World Health Organization list of priority antibiotic resistant pathogens.”

Comment 2: Line 78 – 81 The authors could explain why ESBL-PE colonisation was associated with the SHV enzyme and reasons for more the predomination of the CTM-X enzyme more recently.

Response: The reason for the more recent predominance of CTM-X has been added to the final sentence of the 3rd paragraph of the introduction. It is also explained in the 3rd paragraph of discussion of the paper. 

Correction: 

Lines 83 - 85 “More recent studies suggest that the cefotaxime-hydrolyzing beta-lactamase, (CTX-M) enzyme now predominates because of superior transmission efficiency.” 

Lines 393 - 396 “This is in keeping with the global pattern where CTX-M has become the predominant ESBL-conferring gene in resistant K. pneumoniae and E. coli isolates [4, 21, 27]. It has been suggested that the CTX-M plasmid has adapted to K. pneumoniae resulting in better transmission efficiency [4, 27].”

Comment 3: Line 125. There is no assessment or investigation around whether or not children were transferred from another healthcare facility or convalescence home – is the sample size big enough to look at this?

Response: This was reported in Table 2, item 3. (A foot note has been included in Table 2 to explain this). It was not significant on univariate analysis.

Correction: An addition has been included to the foot note for Table 2 “@Tranfered to RCWMCH refers to admitted from another hospital or a convalescent home.” 

Comment 4: Line 140. It would be useful to know if samples were processed at the GSH laboratory on the day they were received. And if not, how they were stored and if this could influence the results in any way.

Response: Samples were processed at GSH Laboratory within 6 hours of collection. This is stated in the methods section, Lines 142 – 144.

Correction: None required

Results

Comment 5: Figure 1. It would be better to use the word “participant” rather than “patient” when referring to children enrolled in the study?

Response: The word “participant” has been used to replace “patient” in the results and other relevant parts of the manuscript, including figure 1.

Correction: The “patient” has been replaced with “participant” in Lines 123, 124, 125, 126, 128, 135, 141, 195, 270, 273, 275, 280, 281, 283, 285, 305, 307, 347, 348, 374, 382, 383, 422, 423, 444, 458 and figure 1.

Comment: 6. Line 277. It would be more appropriate to describe the participant characteristics before describing prevalence of ESBL-PE or CRE colonisation.

Response: Suggestion has been implemented.

Correction: Participant characteristics (Lines 270 - 278) now appears before prevalence of ESBL-PE or CRE colonisation (Lines 279 - 288).

Comment 7: Table 1. In the text, the authors report that 96 children were colonised by either ESBL-PE or CRE, but in the table it’s indicated that 97 children where colonised? This should be corrected.

Response: In total, 97 participants were colonised by a resistant enterobacterales, 96 by an ESBL-PE and one additional participant by a CRE. ”additional” was added to the sentence, to improve the clarity of the description.

Correction: Lines 280 – 284 “Of 200 participants enrolled, 96 were colonised by at least one ESBL-PE, giving an ESBL-PE colonisation prevalence of 48% (95% CI 40.9-55.2%). Of these, 8 participants were colonised by two different ESBL-PE, prevalence of 4% (95% CI 1.9-7.5%). One additional participant was colonised by a carbapenem-resistant Enterobacter cloacae (E. cloacae) giving a CRE colonisation prevalence of 0.5% (95% CI 0.02–2.4%).”

Comment 8: Table 1. It would be useful to know why the HIV category of 23.5% of children is unknown.

Response: HIV test is not done routinely in all patients admitted into the hospital, especially among patients admitted for surgical procedures.

Correction: A brief explanation included in Lines 274 – 276 “The HIV status in 23.5% (47/200) participants was unknown, as HIV status is not determined routinely in all patients admitted into the hospital.”

Comment 9: Line 305. It would be useful to mention in this line, the number of children that had known previous antibiotic use and were colonisation by ESBL-PE.

Response: This is reflected in lines 313 -315. A breakdown of previous antibiotic exposure by colonisation status is also included in table 2

Correction: None. Explanation above.

Comment 10: Line 336. Again, here the number of colonised children is listed as 97 whereas in other places it’s 96? The manuscript should be checked throughout for consistency.

Response: Explanation as in response to Comment 7 above.

Correction: None required.

Discussion

Comment 11: It might be interesting to discuss in more detail the kinds of studies that could be conducted to build upon this one and if and how whole-genome sequencing could contribute.

Response: Suggestion has been included in the conclusion.

Correction: Lines 444 – 451 “Further research, adequately funded to include more participants, is required to provide robust colonisation estimates, explore colonisation prevalence changes over time, quantify the extent of community versus hospital acquisition, determine the pattern of acquisition of colonisation in the course of admission and relatedness of the isolates, and whether improved infection control practice can moderate colonisation rates. Whole genome sequencing will additionally help interrogate other resistance conferring genes not identified by PCR technology among the ESBL-PE and CRE isolates, especially among those where no resistance conferring gene was found.”

Reviewer 4

Comment 1: Line 82: Twelve studies are cited with prevalence rates. Were all the studies relevant to describing colonization rates and what proportion included the neonatal population? 

Response: All the studies reported prevalence of CRE colonisation in children, some did not present aggregated data to ascertain prevalence among neonates. Eight of the 12 studies specifically reported inclusion of neonatal population, but specific numbers involved were not reported in all.

Correction: None required.

Results:

Comment 2: Neonates formed a small percentage of the study population in the study. Was this due to the sampling method? 

Response: Most neonates are admitted into another secondary level hospital. This explains the low number of neonates enrolled into the study.

Correction: None.

Comment 3: Were there any specific risk factors identified in those neonates who were colonised with ESBL-PE?

Response: This was not done due to the very small number of neonates, which precluded specific sub-analysis among neonates.

Correction: None.

Discussion:

Comment 4: Since the high prevalence of ESBL-PE colonisation in the study population was identified, were any infection prevention and control interventions introduced or enhanced at the hospital to prevent transmission to other patients? 

Response: The results of the study was communicated to the parents/legal guardian and managing teams, and appropriate IPC measures were emphasised.

Correction: None required.

Minor points:

Comment 5: Line 78: the spacing of “new born” 

Response: Correction implemented.

Correction: Line 82 “…protection against ESBL-PE colonisation in the newborn period.”

Comment 6: Line 273: Spelling of “cloaca”

Response: Spelling has been corrected.

Correction: Line 282 – 283 “One participant was colonised by a carbapenem-resistant Enterobacter cloacae (E. cloacae) giving…”

Comment 7: Line 406: use italics for: E. coli

Response: E. coli has been rendered in italics.

Correction: Lines 415 – 416 “The high proportion of E. coli (91.6%) susceptible to both piperacillin-tazobactam and amikacin suggest this may be a consideration in BSI caused by ESBL-PE E. coli.”

Comment 8: Line 400-405: The reference to the study is not cited

Response: The reference has been included. 

Correction: Lines 413 – 415 “…, suggesting that piperacillin-tazobactam alone may no longer be the recommended definitive therapy in this context [56].”

In the references section:

Lines 695 – 699 “56. Harris PNA, Tambyah PA, Lye DC, Mo Y, Lee TH, Yilmaz M, et al. Effect of Piperacillin-Tazobactam vs Meropenem on 30-Day Mortality for Patients With E coli or Klebsiella pneumoniae Bloodstream Infection and Ceftriaxone Resistance: A Randomized Clinical Trial. Jama. 2018;320(10):984-94. Epub 2018/09/13. doi: 10.1001/jama.2018.12163. PubMed PMID: 30208454; PubMed Central PMCID: PMCPmc6143100.”

---

## [Editor Report · Decision Letter 1]

21 Oct 2020

Colonisation with extended spectrum beta-lactamase-producing and carbapenem-resistant Enterobacterales in children admitted to a paediatric referral hospital in South Africa

PONE-D-20-22787R1

Dear Dr. Ogunbosi,

We’re pleased to inform you that your manuscript has been judged scientifically suitable for publication and will be formally accepted for publication once it meets all outstanding technical requirements.

Kind regards,

Mehreen Arshad, M.D.

Academic Editor

PLOS ONE
---

## [Editor Report · Acceptance letter]

27 Oct 2020

PONE-D-20-22787R1 

Colonisation with extended spectrum beta-lactamase-producing and carbapenem-resistant Enterobacterales in children admitted to a paediatric referral hospital in South Africa 

Dear Dr. Ogunbosi:

I'm pleased to inform you that your manuscript has been deemed suitable for publication in PLOS ONE. Congratulations! Your manuscript is now with our production department. 

Kind regards, 

on behalf of

Dr. Mehreen Arshad 

Academic Editor

PLOS ONE